# Rac1 Promotes Cell Motility by Controlling Cell Mechanics in Human Glioblastoma

**DOI:** 10.3390/cancers12061667

**Published:** 2020-06-23

**Authors:** Jing Xu, Nicola Galvanetto, Jihua Nie, Yili Yang, Vincent Torre

**Affiliations:** 1International School for Advanced Studies (SISSA), 34136 Trieste, Italy; jixu@sissa.it (J.X.); nicola.galvanetto@sissa.it (N.G.); jnie@sissa.it (J.N.); 2Joint Laboratory of Biophysics and Translational Medicine, Suzhou Institute of Systems Medicine (ISM)- International School for Advanced Studies (SISSA), Suzhou 215123, China; 3School of Radiation Medicine and Protection, State Key Laboratory of Radiation Medicine and Protection Medical College of Soochow University, Suzhou 215123, China; 4Cixi Institute of Biomedical Engineering, Ningbo Institute of Materials Technology and Engineering Chinese Academy of Sciences, Ningbo 315201, China

**Keywords:** Rac1, cell motility, cell mechanics, cytoskeleton, cell adhesion

## Abstract

The failure of existing therapies in treating human glioblastoma (GBM) mostly is due to the ability of GBM to infiltrate into healthy regions of the brain; however, the relationship between cell motility and cell mechanics is not well understood. Here, we used atomic force microscopy (AFM), live-cell imaging, and biochemical tools to study the connection between motility and mechanics in human GBM cells. It was found thatRac1 inactivation by genomic silencing and inhibition with EHT 1864 reduced cell motility, inhibited cell ruffles, and disrupted the dynamics of cytoskeleton organization and cell adhesion. These changes were correlated with abnormal localization of myosin IIa and a rapid suppression of the phosphorylation of Erk1/2. At the same time, AFM measurements of the GBM cells revealed a significant increase in cell elasticity and viscosity following Rac1 inhibition. These results indicate that mechanical properties profoundly affect cell motility and may play an important role in the infiltration of GBM. It is conceivable that the mechanical characters might be used as markers for further surgical and therapeutical interventions.

## 1. Introduction

Glioblastoma (GBM; World Health Organization grade IV glioma) is the most common and incurable primary brain tumor. Its infiltration ability is largely responsible for the failure of existing therapies. Despite significant progress in developing targeted agents and immunotherapies, the prognosis of patients with GBM has not been improved markedly. Current GBM treatments focus on neurosurgical resection followed by radiation and chemotherapy. Unfortunately, the median survival time of patients with GBM is less than fifteen months [1,2]. Novel diagnostic strategies and tools to rapidly determine the response of individual GBM to drugs are highly desirable to improve patient survival.

Rho GTPases are known to be at the basis of infiltration, regulating the dynamics of the cytoskeleton and cell adhesion and providing a fine adjustment of cell movements [3,4]. In migrating cells, Rac1 is well known for its function in lamellipodia formation, which serves as a major driving force of cell movement [5,6]. Rac1 mediates actin polymerization though WAVE family proteins and activates the Arp2/3 complex for branched actin filament formation in lamellipodia [7,8,9]. Rac1 is also involved in fast plasma membrane movements, such as cell membrane ruffling and vesicular transport [6,10]. Interestingly, deregulation of Rac1 has been recently found in a variety of cancers [11,12,13]. It has also been shown that increased Rac1 activation promotes cancer cell proliferation, migration, and metastasis [3,14,15]. However, how aberrant Rac1 activation plays a role during GBM migration is still not fully understood.

Similar to most other Rho GTPases, Rac1 usually switches between two conformations with GDP and GTP and mediates activation. Their intrinsic GTPase activity coupled with GAPs stimulates GTP hydrolysis and leads to the inactive state, which is then extracted from the cell membrane under the influence of guanine nucleotide exchange inhibitors (GDIs) [16]. The small molecule EHT 1864 binds to Rac1 with a high affinity and blocks Rac1 interaction with its effectors (guanine nucleotide-associated effectors to Rac1) both in vivo and in vitro [17]. Therefore, EHT 1864 is a powerful probe to evaluate the function of Rac1.

The mechanical properties of single cells are inextricably linked to their intracellular components. Cells usually change their mechanical properties during development and disease. Most malignant cells have been reported to be softer than normal tissue cells [18,19]. Drugs that affect the cytoskeleton or cell architecture are often used to treat cancer or other cellular diseases [20,21,22]. Among these drugs, paclitaxel, dexamethasone, and daunorubicin modify the mechanical phenotypes of cells [23,24]. However, further studies are still needed to understand how these intracellular components, such as the cytoskeleton and cell architecture, contribute to cell mechanical properties. 

In this study, we blocked Rac1 activation by gene knockdown and Rac-specific inhibitor EHT 1864 to investigate how and to what extent Rac1 mediates GBM cell motility. We focused our attention on Rac1 regulating plasma membrane movements and the dynamics of the cytoskeleton and cell adhesions during GBM migration. We also examined which downstream effectors of Rac1 are important in GBM motility. AFM was used to explore the relationship between cell infiltration and cell mechanics in the presence or absence of Rac1 inhibition.

## 2. Results

### 2.1. Rac1 Is Essential for GBM Motility

Rac1 is the Rho GTPase family member that is mostly expressed in gliomas, and its level of expression correlates with patient survival outcome (Appendix A). In this study, three types of typical GBM cell lines, U87 (glioblastoma multiforme grade IV), U251 (glioblastoma-astrocytoma grade III–IV), and T98G (glioblastoma multiforme grade IV), were used to study motility and associated changes. GBM cells display strong invasion properties, and these cells efficiently migrate through narrow pores (~8 μm) or invade into a three-dimensional Matrigel matrix. To investigate whether Rac1 is required for GBM cell invasion, small interfering RNA (siRNA) specifically against Rac1 was used to knockdown its expression in U87, U251, and T98G cells (Figure 1a and Appendix A). Knockdown of Rac1 prevented the invasive phenotype of all these GBM cells (in U87 cells, 37.5 ± 13% in siRNA#1 group and 56.2 ± 11% in siRNA#2 group; in U251 cells 19.4 ± 16% in siRNA#1 group and 34 ± 17% in siRNA#2 group; and in T98G cells 65.7 ± 26% in siRNA#1 group and 66.8 ± 20% in siRNA#2 group passed through ~8μm pores coated with Matrigel, Figure 1c,d). The small EHT 1864 (M.W. 581.47) blocks the binding of guanine nucleotide-associated effectors to Rac1 [17]. To further detect the role of Rac1 in GBM motility, a concentration of 10 μM EHT 1864 was chosen because it had a limited effect on GBM cell viability (Appendix A). The levels of active Rac1 were assessed using an active GTPase pull-down assay. As shown by immunoblotting measurements in GBM cells, incubation with EHT 1864 for 1 or 2 h reduced the levels of Rac1-GTP (Figure 1b). Upon inhibition, the invasion ability of these cells was reduced (21 ± 13% of U87 cells, 50 ± 13% of U251 cells, and 21 ± 23% of T98G cells passed through the pores, Figure 1e,f).

We next tested the role of Rac1 in the motility of GBM cells using live cell imaging. A stable mCherry-U87 cell line was established by using LV_Pgk1p-mCherry to visualize GBM movements. U87 cells usually quickly change their shape during movement (Figure 2a upper panel, Appendix A). Upon Rac1 inhibition, U87 cells became rounder (Figure 2a lower panel, Appendix A) and reduced their spreading area (Figure 2b). Trajectories of individual cells were used to quantify motility differences following EHT 1864 treatment (Figure 2c,d). We verified that Rac1 inhibition of GBM significantly reduced the velocity of U87 cells (Figure 2e).

### 2.2. Rac1 Signaling Regulates Myosin IIa Localization

In the leading edge, cells quickly formed membrane ruffles and protrusions for cell movement. U87 and U251 cells usually exhibited epithelial-like morphology and formed lamellipodia in front of cell. However, knockdown of Rac1 led to cell morphological changes and the formation of long protrusions (Figure 3a,b and Figure 4). Inhibition ofRac1 signaling by EHT 1864 also showed that Rac1 was involved in the formation of membrane ruffles and protrusions (Appendix A), polymerization of stress actin fibers (Appendix A), and tubulin (Appendix A) in lamellipodia. 

Non-muscle myosin II is an actin-binding protein and plays an important role in cell contraction during cell migration. Rac1 activation allows GBM cells to change their shape for their movement (Appendix A). We were interested whether Rac1 signaling regulates myosin II during cell movement. Immunoblotting analysis indicated that, following Rac1 knockdown or inhibition, myosin IIa phosphorylation levels did not significantly change (Appendix A). However, we found that myosin IIa usually represented a gradient from the cell rear to the leading edge in normal U87 and U251 cells. After Rac1 depletion, this gradient changed, and myosin IIa also exhibited a significant degree in the newly formed, long protrusions (Figure 3a,b). Furthermore, myosin IIa rarely localized in the leading region according to EHT 1864 inhibition (Figure 3c, Appendix A).

### 2.3. Rac1 Signaling in Cell Adhesion Formation

Inhibition of Rac1 activity disrupted lamella formation and induced a reduction in cell motility. We next analyzed the formation of cell adhesions in GBM. Cell adhesions form at the leading edge of protrusions and disassemble at both the leading edge and at the rear of the cell. These events are at the basis of cell migration. Depletion of Rac1 in U87 and U251 allowed cells to form long, thin cellular protrusions, where cells were rich in actin filaments and exhibited lack of Paxillin spots (Figure 4a,b). To determine whether Rac1 activity is involved in the rapid assembly and disassembly of adhesions, U87 cells expressing mCherry-Lifeact-7 and Paxillin-GFP were used. In this way, we monitored the dynamics of adhesions and actin organization using live cell imaging (Figure 5a, Appendix A). Recordings lasted approximately 15 min, and one frame was digitized every 10 s. Analysis of these recordings showed four distinct adhesion types: assembly type, in which a single assembly event was observed within 15 min; disassembly type in a single disassembly event was seen; stable type, in which the adhesion spot remained stable; and turnover type, which showed both assembly and disassembly events (Figure 5f). Our data show that in U87 GBM cells, after adding EHT 1864 for 2 h, more adhesion spots were stable, while both disassembly and turnover types were less frequent (Figure 5a,f). Although the number of adhesions (Figure 5b), the area of adhesions (Figure 5c) and the assembly rate (Figure 5d) was less affected by the addition of EHT 1864, the disassembly rate (Figure 5e) were less affected by the addition of EHT 1864, the disassembly rate (Figure 5e) was dramatically decreased, concomitant with a blockage of actin filament collapse (Figure 5a). Furthermore, the lifetime of turnover adhesions did not exhibit a detectable difference between each group (Figure 5i); however, both the assembly and disassembly rates of these adhesions were significantly slower than the rates in control conditions (Figure 5g–j), which explains why the portion of turnover type was low following Rac1 inhibition.

### 2.4. Rac1 Activates Erk to Mediate Cell Adhesion Dynamics

The mitogen-activated protein kinase (MAP kinase)/Erk cascade promotes Paxillin phosphorylation [25]. Inhibition of the Erk1/2 upstream regulator MEK1/2 constitutively abolishes active Rac1-induced EMT in ovarian cancer cells [26]. Our data showed that inhibition of Rac1 contributes to impaired Paxillin disassembly and turnover of cell adhesions. We then investigated whether Erk1/2 is involved in Rac1 signaling that regulates the dynamics of cell adhesion. Immunoblotting analysis indicated that, following Rac1 inhibition, Erk1/2 phosphorylation levels were diminished within 30 min and up to 120 min in a time-dependent manner after EHT 1864 was added (Figure 6). These results show that the Erk1/2 pathway may be involved in Rac1 regulation of GBM cell motility.

### 2.5. Characterization of the Mechanical Properties in Response to Rac1 Inhibition

Several works have shown that the elastic properties of cancer cells are correlated with malignant transformation [18,26,27]. The cell cytoskeleton, especially the actin cytoskeleton, is the key regulator for maintaining cell shape and mechanics [28,29]. Changes in cytoskeletal organization and cell shape during migration were observed in response to Rac1 inhibition (Figure 2b, Appendix A). We further decided to detect the elastic properties of GBM cells, and single-cell force spectroscopy (SCFS) measurements were performed using AFM (Figure 7a,b). Cell elasticity values were obtained by fitting the approaching part of the recorded F–D curves using the Hertz–Sneddon model and calculating the Young’s modulus (E) of each cell (Figure 7c,d). Quantitative analysis showed that, following Rac1 inhibition, the Young’s modulus of GBM cells increased from 0.84 ± 0.53 kPa to 1.12 ± 0.70 kPa (Figure 7e). Cellular viscosity can be used to characterize the fluidity or ease of movement inside cells, which is an important mechanical property of live cells. We hypothesized that, following the inhibition of Rac1, the viscosity of GBM could be changed. The viscosity of GBM cells was obtained following the procedure described by Achu Yango et al. using AFM (Figure 7f–i) [30]. Consistent with the elasticity changes, the viscosity of live GBM cells also increased from 193.59 ± 152.16 Pa × s to 257.15 ± 187.82 Pa × s following Rac1 inhibition (Figure 7j). The increased viscosity observed here is consistent with the view that, following inhibition of Rac1, the whole cell interior reorganizes and is less fluid and more rigid.

## 3. Discussion

In the present study, we confirmed that Rac1 is involved in GBM infiltration. We showed that Rac1 depletion or inhibition leads to the following: I. GBM cells dramatically reduce their invasion ability; II. GBM cells change their morphology accompanied with a reorganization of the cell cytoskeleton, abnormal myosin IIa location, and cell adhesion formation; III. there is a rapid suppression of Erk1/2 phosphorylation according to EHT 1864 inhibition; and IV. this was accompanied by an increase in the rigidity and viscosity of GBM cells. 

Upregulation of Rac1 has been recently reported in various kinds of cancers and is correlated with poor overall survival of patients with glioma (Appendix A). Recent works report that Rac1 plays a very important role in cancer cell proliferation and survival. In our study, knockdown of the expression of Rac1 in U87, U251, and T98G cells did not significantly affect cell viability (Appendix A), which indicates that Rac1 promotes glioma progression mainly by another way. Regarding GBM, in particular, U87 cells were observed in a co-culture condition with healthy neurons and glia, and visual inspection revealed a remarkable feature: GBM cells had a much higher motility than healthy neurons and glial cells [31]. Activated Rac1 induced the formation of membrane ruffles and cell protrusion at the leading edge of migrating cells [11,32]. Knockdown of Rac1 in both U87 and U251 showed that cell lacked lamellipodium, instead of forming long, thin cell protrusions (Figure 3a,b and Figure 4). In the presence of Rac1 inhibitor EHT 1864, cells showed frequent membrane protrusions but without ruffle activity (Appendix A). Activated Rac1 allows WAVE to stimulate Arp2/3-actin polymerization activity in the lamellipodium [33]. Abnormal ruffles and shrinkage of the cell body suggest that Rac1 inactivation causes a reorganization of actin in GBM cells. Using SiR-actin and SiR-tubulin to label cytoskeleton filaments, we found that Rac1 inhibition caused defects in actin organization and tubulin direction (Appendix A).

Impaired disassembly of cell adhesion affects cell migration in murine embryonic fibroblasts (MEFs); however, constitutive activation of Rac1 shows no detectable effect on adhesion disassembly [25]. Here, we found that Rac1 depletion in GBM cells promoted the formation of long, thin protrusion sand lack of Paxillin spots (Figure 4a,b). Furthermore, inhibition of Rac1 by EHT 1864 disrupted the disassembly of cell adhesion concomitant with defects in the actin web and switched the proportion of stable cell adhesion complex (Figure 5a and Figure 4e,f, Appendix A). These results demonstrate that Rac1 plays different roles in different cell types in regulating cell adhesion formation. In migrating cells, non-muscle myosin II can bind to actin and lead to cell contraction that increases cell membrane protrusion at the leading edge and inhibits protrusion at the rear of the cell [34,35]. Our results indicated that there were no detectable changes in myosin IIa activity (Appendix A). Myosin IIa exhibited a high degree of long, thin protrusions of Rac1-depleted GBM cells, which maybe the reason for these cells losing their polarity for fast movement. Myosin IIa rarely emerged at the leading stage according to EHT 1864 inhibition (Figure 3, Appendix A). This demonstrates a lack of forces that induce contraction and breakage of cell–matrix adhesions at the rear mainly due to the abnormal distribution of myosin IIa. Donna J. Webb and colleagues demonstrated that Erk and MLCK activated the phosphorylation state of Paxillin, which further regulates adhesion disassembly [25]. Here, we found a rapid decrease in the phosphorylation level of Erk1/2 (Figure 6), suggesting that Rac1 also regulates the dynamics of cell adhesion through the Erk signaling pathway.

Previous studies have reported that cancer cells are softer than normal cells [18,19], which allows metastatic cells to squeeze their shape through the extracellular matrix. The mechanical properties have also been used to evaluate cancer progression and the reaction to drugs [27]. The mechanical properties depend primarily on the cytoskeleton, and it is expected that the decreased value of cell elasticity (Figure 7c–e) caused by Rac1 inhibition is associated with the disruption of cytoskeleton organization inside GBM cells (Appendix A). Viscosity is a measure of fluid resistance, and the inhibition of Rac1 causes an increase in viscosity, i.e., a decrease in the fluidity of the cell (Figure 7f–j). These results indicate that cell mechanical properties change according to pharmacologic treatment, and these changes are associated with cell motility in GBM cells.

## 4. Materials and Methods 

### 4.1. Cell Lines

Human GBM cell lines U87, U251, T98G (all from ATCC), and U87 stably transfected with LV_Pgk1p-mCherry were cultured at 37 °C, 5% CO_2_ in DMEM medium supplemented with 10% FBS and penicillin/streptomycin. Cell cultures were routinely subcultured every two days by trypsinization using standard procedures. U87 stably transfected with LV_Pgk1p-mCherrywere provided by the laboratory of Prof. Antonello Mallamaci from the International School of Advanced Studies.

### 4.2. Gene Silencing

siRNAs against Rac1 and the negative control (NC) were purchased from Shanghai GenePharma Co., Ltd (Shanghai, China). Target sequences for siRac1 were as follows: siRac1#1, 5′-CUACUGUCUUUGACAAUUATT-3′ and 5′-UAAUUGUCAAAGACAGUAGTT-3′; siRac1#2, 5′-GAGUCCUGCAUCAUUUGAATT-3′ and 5′-UUCAAAUGAUGCAGGACUCTT-3′.

siRNAs were transiently transfected into U87, U251, or T98G cells by Lipofectamine^TM^ RNAiMAX (Invitrogen, Carlsbad, CA, USA) according to the manufacturer’s instructions.

### 4.3. Immunofluorescent Staining and RT-PCR

Cells were fixed with 4% PFA for 20 min and permeabilized with 0.1% TritonX-100 for 20 min. Cells were then incubated with primary antibody and secondary antibody. The following primary antibodies were used: anti-myosin IIa (1:200, Cell Signaling Technology, Danvers, MA, USA) and anti-Paxillin (1:200, Abcam, Cambridge, UK). Phalloidin (1:2000, Invitrogen) was used to detect F-actin.

Total RNA was extracted using Trizol^®^ (Invitrogen), according to the manufacturer’s instructions, and then was reverse-transcribed by PrimeScript^TM^ reagent kit (TaKaRa, Kusatsu, Japan). PCR reaction was done using SUBR^®^ Premix Ex Taq^TM^ (TaKaRa). RAC1 primer sequence was 5′-AAGCTGACTCCCATCACCTATCCG-3′ and 5′-CGAGGGGCTGAGACATTTACAACA-3′. 

### 4.4. Cell Proliferation Assay

A total of 5 × 10^3^ cells were plated in 96-well plates. After 12 h culture, cells were transfected with Rac1-siRNA and NC-siRNA and cultured for another 48 h. Cell growth was measured by a colorimetric CCK-8 assay (Sigma-Aldrich Co, St. Louis, MO, USA).

A total of 2.5 × 10^4^ cells were plated in 96-well plates. After 12 h culture, cells were treated with different concentrations of EHT 1864 2HCL (Sellekchem, Houston, TX, USA) for another 24 h, and then CCK-8 measurement followed.

### 4.5. Cell Transfection and Live Cell Imaging

Cells were plated at a density of 1 × 10^5^ cells/mL into 35 mm confocal dishes and incubated overnight. SiR-actin and SiR-tubulin (cytoskeleton, Denver, CO, USA) staining were used to detect the dynamics of actin and tubulin cytoskeleton following the protocol provided by the manufacturer. Transient transfections of mCherry-Lifeact-7 and paxillin-GFP were carried out with Lipofectamine™ 2000 (Invitrogen) using the protocol provided by the manufacturer. Twenty-four hours after transfection, cells were used for the live cell imaging assay. mCherry-Lifeact-7 was a gift from Michael Davidson (Addgene plasmid # 54491; http://n2t.net/addgene:54491; RRID:Addgene_54491).

For the live cell imaging assay, cells were maintained at 37 C, 5% CO_2_ in chambers in culture medium on an inverted microscope (Nikon, Tokyo metropolis, Japan) with a motorized stage (Prior Scientific, Cambridge, UK) controlled by Simple PCI software (Compix). Stable U87-mCherry cells were used to detect the cell shape, and velocity changes with and without EHT 1864 and time-lapse image series were acquired at 1 min intervals using a 20 × 1.4 NA objective lens (Nikon). SiR-actin, SiR-tubulin, mCherry-Lifeact-7, and Paxillin-GFP fluorescence time-lapse image series were acquired at 1 s to 2 min intervals using a 40 × or 60 × 1.4 NA objective lens (Nikon). Cells were treated with 10 μM EHT 1864 to detect the role in cytoskeleton organization and dynamics of cell adhesions.

### 4.6. Tracking Assays and Data Analysis

Tracking of cell migration was performed over a period of 4 h. Individual cells were tracked by repeated selection of cells in movie frames and manual tracing of migration pathways with Image J software.

Individual cell adhesions were tracked similar to the cell tracking procedure. We then fit the curves of ‘adhesion assembly’ or ‘adhesion disassembly’ by Origin software using Slogistic1 function, and we obtained the *k*/-*k* value as the rate of assembly or disassembly. To calculate the lifetime of turnover adhesions, we separated the curves into two parts and fit them individually.

### 4.7. Active Rac1 Pull-Down Assay and Western Blot

Cells were plated at a density of 2 × 10^6^ cells/well in 10 cm and grown to 70–80% confluence. The cells were then incubated with 10 μM EHT 1864 for 15, 30, 60, and 120 min. Cells were washed with cold PBS and lysed in IP lysis buffer (20 mMTris pH 7.4, 150 mM NaCl, 5 mM MgCl_2_, 0.5% NP-40, 10% glycerol, pH 7.4) supplemented with a protease inhibitor cocktail (Sigma-Aldrich Co, St. Louis, MO, USA). After normalization, lysates were incubated with 20 μg PAK-PBD agarose for 60 min at 4 ℃ with rotation, and then the beads were collected for Western Blotting analysis.

Equal amounts of total protein were boiled for 5 min in 5× sample buffer and fractionated by 12% SDS-PAGE. Samples were then transferred to PVDF Membranes (Millipore, Danvers, MA, USA). Immunoblots were detected using the ECL System (Millipore) with horseradish peroxidase-conjugated secondary antibodies (Sigma-Aldrich Co, St. Louis, MO, USA). The following first antibodies were used: anti-Rac1 (Abcam, Cambridge, UK), anti-p-Erk1/2 (Cell Signaling Technology, Danvers, MA, USA), and anti-Erk1/2 (Cell Signaling Technology, Danvers, MA, USA).

### 4.8. Transwell Assay

To detect cell invasion, transwell chambers were pre-coated with 200 μL of a 0.8 mg/mL Matrigel suspension and dried at 37 ℃. Cells were starved overnight in DMEM medium, 5 × 10^5^ cells in DMEM medium were added to the top chambers of 12-well transwell plates (Nunc; 8 μm pore size), and 15% FBS DMEM medium was added to the bottom chambers. After 48 h incubation, top (non-migrating) cells were removed, and bottom (migrating) cells were fixed with 4% PFA and stained with 5% crystal violet.

### 4.9. Force Spectroscopy of Living Cells and Data Analysis

Single-cell force spectroscopy (SCFS) measurements were performed by using a commercial AFM (JPK Instruments, Berlin, Germany) mounted on top of an Axiovert 200 inverted microscope (Olympus, Japan). We used rectangular low-stress SiN cantilevers terminated with a silicon pyramidal tip (APPNANO, Mountain View, CA, USA). Those cantilevers are characterized by a nominal spring constant *k* = 0.0084 N/m; the tip height was 4–6 μm, while the radius of curvature at the tip apex was <25 nm. The cells were incubated 37 °C using a temperature-controlled BioCell chamber (JPK Instruments, Berlin, Germany) during the measurements. Cell elasticity values were obtained by fitting the approaching part of the recorded F–D curves using the Hertz–Sneddon model and calculating the Young’s modulus (E) of each cell through JPK software. Cell viscosity measurements were obtained as previously described [36]. Briefly, the z voltage was kept constant for 2 s after approaching the sample, the z height was increased by a small amount (0.5 nN) and kept for 1 s, and then it was withdrawn again after 1 s.

### 4.10. Statistical Analysis

Differences between groups were assessed by Student’s *t* test, one-way ANOVA, or two-way ANOVA test. GraphPad Prism was used for all statistical analyses. The differences of RT-PCR, WB, and CCK-8 were evaluated by two-way ANOVA. The differences in cell mobility, mechanics, and adhesion dynamics were evaluated by a Student’s *t* test followed by a Mann–Whitney test. The difference of cell spreading area was evaluated by one-way ANOVA followed by a Kruskal–Wallis test. The results are presented as the mean ± SEM of at least three independent experiments.

## 5. Conclusions

Our study demonstrates that there is a close relationship between cell mechanics and invasiveness of GBM cells. In GBM cells, downregulation of Rac1 is associated with low cell motility by regulating the dynamics of the cytoskeleton and cell adhesion. Rac1 activation also adjusts the rigidity and viscosity of GBM cells. It takes only a few minutes to measure the single-cell mechanics, which is a much shorter time compared to the typical original assays. Here, we suggest that cell mechanics can be used as a novel and convenient strategy for surgical and therapeutical interventions in the future.

## Figures and Tables

**Figure 1 cancers-12-01667-f001:**
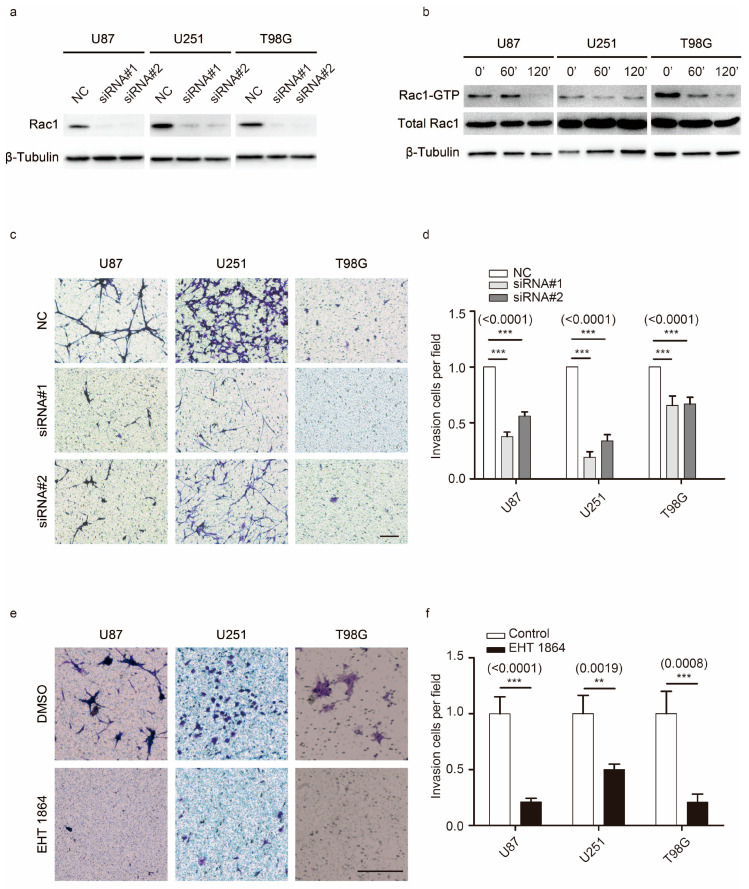
Rac1 is essential for glioblastoma (GBM) invasion. (**a**) Western blot analysis for Rac1 expression in U87, U251, and T98G cells 48 h after treating with siRNAs against Rac1 and negative control (NC).(**b**) The levels of Rac1-GTP (upper panel) with or without EHT 1864 treatment for 1 h or 2 h were concentrated by PAK-GST pull down assay and detected by Rac1 antibody. The expressions of total Rac1 protein (middle panel) and β-tubulin were also detected. (**c**,**e**) Invasion trans well assays were carried out to detect the invasion ability of U87, U251, and T98G cells uponRac1 knockdown (**c**) and 10 μM EHT 1864 treatments (**e**). (**d**,**f**) Cell invasion ability was calculated by counting the number of cells per field. ***: *p* < 0.001, **: *p* < 0.01. Scale bar: 1000 μm. Uncropped blots are shown in Appendix A.

**Figure 2 cancers-12-01667-f002:**
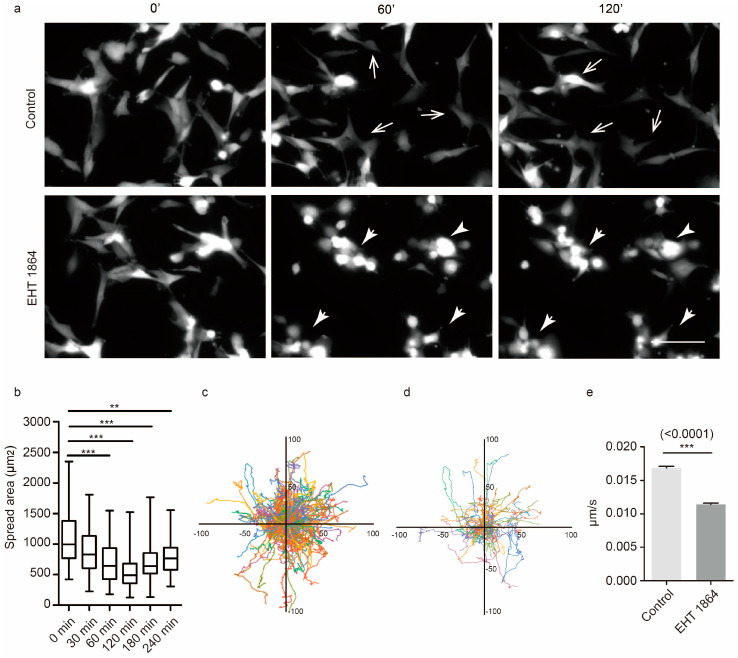
Rac1 activity affects random movement. (**a**) Time-lapse images of U87-mCherry cells. After 4 h recording, cells were incubated with EHT 1864 and recorded for another 4 h. Open arrow indicates cells that rapidly move, and the solid arrow indicates cells that slowly move. (**b**) U87-mCherrycell spreading areas were quantified using the ImageJ program (NIH) after EHT 1864 added for 30, 60, 120, 180, and 240 min. (**c,d**) Cell trajectories of normal U87 cells (**c**) and EHT 1864-treated U87 cells (**d**) for 4 h; each color represents the trajectory of an individual cell, and the starting positions of each cell were registered to the center of the plot. (**e**) The mean velocity of U87-mCherry cells was recorded for 4 h and analyzed using the ImageJ program (NIH). Recordings of U87-mCherrycell movement are shown in Appendix A. Cell number: 277 cells in control group and 241 cells in EHT 1864 treated group. ***: *p* < 0.001, **: *p* < 0.01. Scale bar: 100 μm.

**Figure 3 cancers-12-01667-f003:**
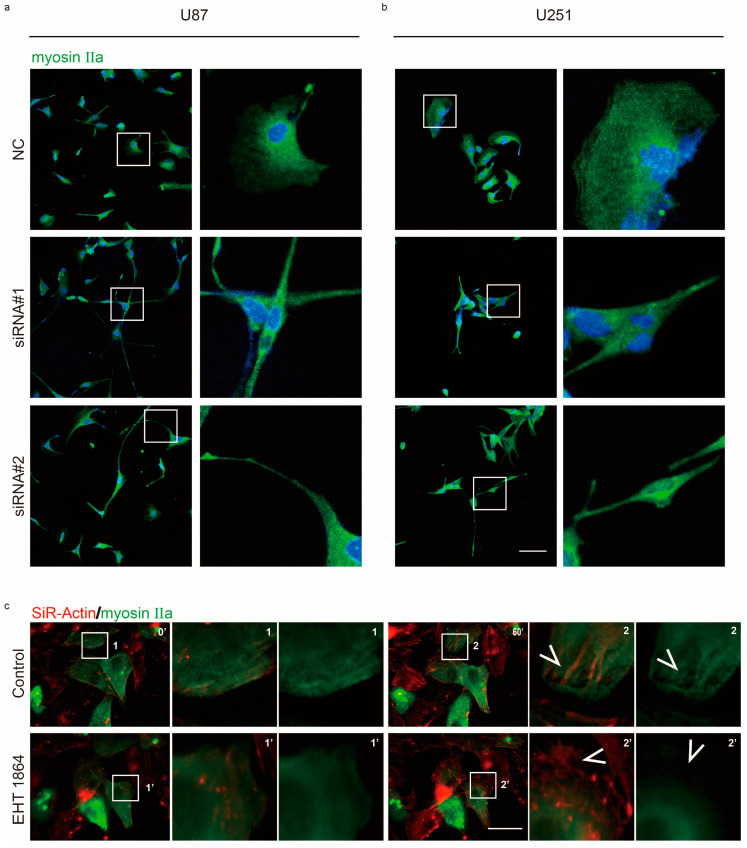
Rac1 regulates myosin IIa localization. (**a,b**) Confocal sections of U87 and U251 cells depleted of Rac1 48 h post-transfection and stained for myosin IIa (green). (**c**) Time-lapse images of SiR-actin staining and myosin IIa-GFP-expressing U87 cells. After 60 min, 10 μm EHT 1864 was added and recorded for another 60 min. Arrows indicate actin fibers and myosin IIa localization in the protrusion. Recordings are shown in Appendix A. Scale bar: 50 μm.

**Figure 4 cancers-12-01667-f004:**
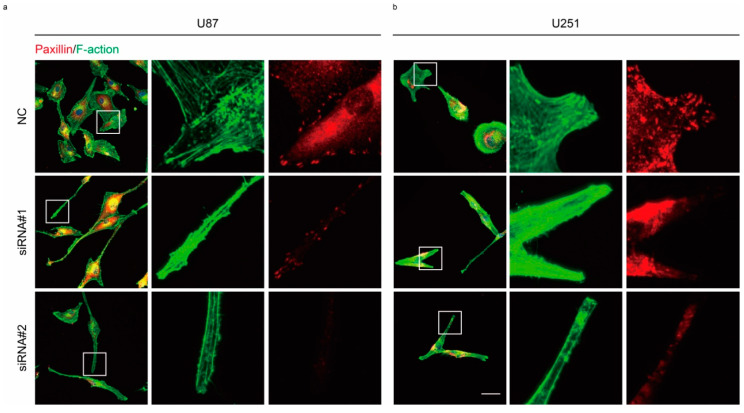
Rac1 is involved in cell adhesion formation. (**a**,**b**) Confocal sections of U87 and U251 cells depleted of Rac1 48 h post-transfection and stained for phalloidin (green) and Paxillin (red). Scale bar: 50 μm.

**Figure 5 cancers-12-01667-f005:**
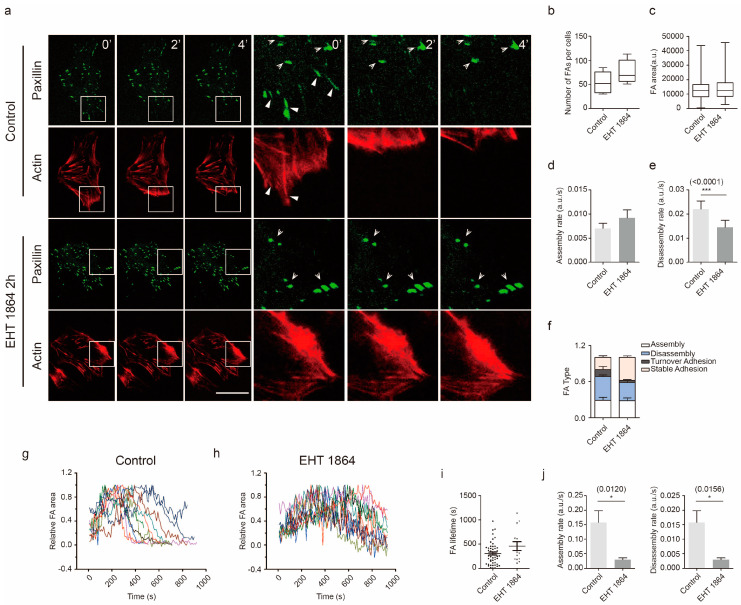
Rac1 signaling controls cell adhesion dynamics. (**a**) U87 transfected with Paxillin (green) and mCherry-Lifeact-7 (red) were imaged by time-lapse microscopy at 10 s intervals for 15 min. Time-lapse images of U87 cells after 1 h of incubation with DMSO or EHT 1864, respectively. Closed arrow indicates disassembly type, and open arrow indicates cell stable adhesion. Histograms of the total cell adhesion number (**b**), adhesion area (**c**), adhesion assembly rate (**d**), and disassembly rate (**e**) and adhesion type proportion ratio in the normal and EHT 1864-treated groups (**f**). (**g–h**) Typical curves of dynamic turnover adhesions in both the control group (**g**) and EHT 1864 group (**h**). (**i**) Lifetime of turnover adhesions in the normal and EHT 1864-treated groups. (**j**) Histograms of the assembly rate and disassembly rate of turnover adhesions in the normal and EHT 1864-treated groups. Recordings are shown in Appendix A. Scale bar: 20 μm. Tracking cell adhesion numbers: 395 in the control group and 407 in EHT 1864-treated group. ***: *p* < 0.001, *: *p* < 0.05.

**Figure 6 cancers-12-01667-f006:**
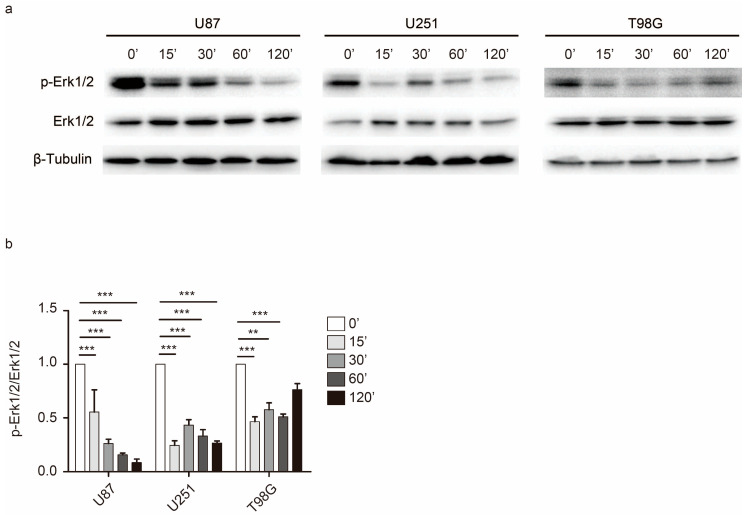
Rac1 regulates cell motility in Erk1/2 dependent way in GBM. (**a**) Western blot analysis of the level of phosphorylation of Erk1/2 in U87, U251 and T98G cells when responding to EHT 1864. (**b**) Histograms of the expression levels of phosphorylated Erk1/2 compared to total Erk1/2 in response to EHT 1864. ***: *p* < 0.001, **: *p* < 0.01. Uncropped blots are shown in Appendix A.

**Figure 7 cancers-12-01667-f007:**
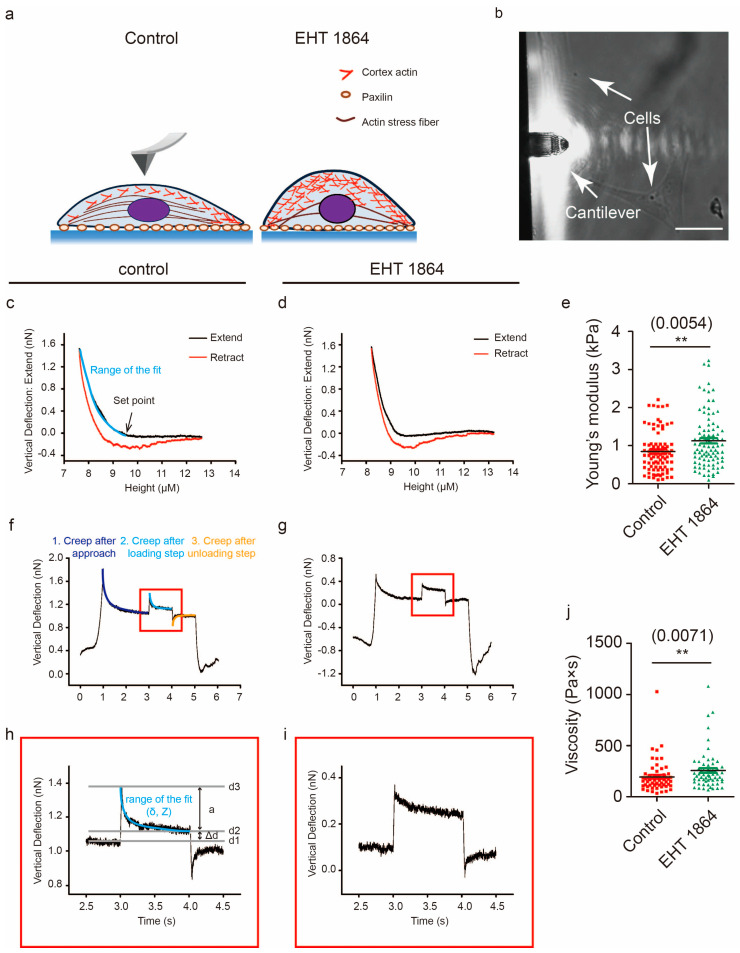
Rac1 activation affects the mechanics of GBM cells. (**a**) Diagrams illustrate the method of application of external force. (**b**) Image showing the AFM detection of the cell mechanical properties of a single living cell. (**c,d**) Exemplary force curve recorded on a U87 cell with (**d**) and without (**c**) EHT 1864 incubation for 1–2 h. The black solid line represents the approach curve, and the Young’s modulus (E) was calculated by fitting the approach curve of each cell. (**e**) Histograms of the associated Young’s modulus E for U87 cells with and without EHT 1864 treatment. (**f–i**) Typical creep curve responses to a cell after applying the z step. The z height is first ramped as in a conventional force curve (approach ramp) and then kept constant for 1 s. (**j**) Histograms of the viscosity value of U87 cells with and without EHT 1864 treatment. Cell number: 95 in control group and 101 in EHT 1864-treated group for Young’s modulus; 64 in control group and 66 in EHT 1864-treated group for viscosity. **: *p* < 0.01; scale bar: 25 μm.

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
