# Peer review of "Rac1 Promotes Cell Motility by Controlling Cell Mechanics in Human Glioblastoma"

_cancers, 2020, doi:10.3390/cancers12061667_

Round 1
Reviewer 1 Report
The revised version of the present manuscript is improved, and my concerns have been largely addressed.
Reviewer 2 Report
The manuscript has been substantially improved and the current results support the authors claims on the role of EHT 1864 in GBM.
This manuscript is a resubmission of an earlier submission. The following is a list of the peer review reports and author responses from that submission.
Round 1
Reviewer 1 Report
* Comments and Suggestions for Authors (will be shown to authors)
The manuscript “Rac1 Promotes Cell Motility by Controlling Cell Mechanics in Human Glioblastoma” by Xu et al. exploreses the effects of EHT1864, a Rac1-inhibitor on cell mechanics (invasion, migration, motility etc.) of GBM cell lines. The manuscript presents some very interesting results. I suggest that the authors should perform some additional experiments to verify the specificity of their assay.
Major Concerns:
- EHT1864 is a known inhibitor of Rac1 activity. However, it is also known that it can also target other Rac family members (Shutes et al, 2007 Pubmed ID:17932039), meaning that its effect may be also due to the inhibition of multiple Rac members in GBM cells. The authors should generate Rac1 KD GBM cell lines and define whether cell mechanics (invasion, migration and motility properties) are altered only by Rac1 inhibition.
- The authors present that EHT1864 induce cytotoxity in a dose dependent manner. Rac1-KD cells should also analysed by CCK-8 to explore whether cytotoxicity is also induced during Rac1 knock down.
- The dose-dependent effect of EHT1864, to invasion and migration properties of GBM cell lines should be also analysed. In Figure 1, 10 μM of EHT1864 seem to totally block both cellular properties. Which is the IC50 of action of EHT1864 on these cellular properties? Additionally, in Figure 1a a marker for total protein (in this case possibly tubulin and not actin) should be also analysed, since there is also a change in total Rac1 in most samples.
Minor concerns:
The manuscript is generally well written. I suggest that the molecule EHT 1864 should be presented uniquely in the text (in many instances is presented as EHT1864, without a space).
Reviewer 2 Report
The manuscript of Xu et al. describes effects of the Rac1/2/3-allosteric inhibitor EHT1864 on glioblastoma cell behavior in vitro. The authors find reduced cell motility and Matrigel invasion upon treating three established glioblastoma cell lines with this compound. In addition, they observe cytoskeletal changes and altered cell adhesion, a change in the subcellular localization of myosin-II, and inhibition of Erk phosphorylation. The authors link these effects with increased cell elasticity and viscosity in the presence of EHT1864, and propose that mechanical properties of cells ‘have a strong correlation with metastasis’ and ‘may provide a novel strategy for future studies on cancer’.
While the manuscript reports interesting observations, a number of questions require clarification before publication.
Major concerns:
EHT1864 is a Rac family inhibitor, yet the authors refer to it as a Rac1-selective inhibitor throughout the manuscript. Are Rac2 and -3 expressed in the investigated glioblastoma cell lines, and do they contribute to the observed effects?
My major concern is the selectivity of the employed EHT1864 inhibitor. Since all results are interpreted as being on-target (i.e., Rac1-mediated), it is critical to document that EHT1864 indeed targets Rac1 in the investigated cell lines at the employed concentrations. Are similar effects (cell migration, invasion, cytoskeletal effects and cell mechanics) seen upon Rac1 siRNA (or CRISPR-Cas9-mediated Rac1 knockout), and can the EHT1864 effects be competed by Rac1 overexpression? How can it be excluded that the observed effects are not off-target? The possibility that EHT1864 inhibits Pak1/2 in a Rac-independent fashion should be considered, as has been reported previously (see Dütting et al., J Thromb Haemost. 2015). Have the authors investigated the possibility that EHT1864 may act by directly inhibiting Erk1/2, thereby causing some of the observed effects?
General points:
All Western blots should be densitometrically quantified (P-myosin-II relative to total myosin-II, normalized to ß-tubulin as a loading control; P-Erk or P-Akt relative to total Erk or Akt normalized to loading control etc.). Are any of the observed changes in Western blot analyses statistically significant after quantification?
All micrographs should include scale bars, and the text/legend should indicate how many cells in how many independent experiments were scored and found to show the reported effects.
Please indicate in all the figures whether data are means +/- standard error or standard error of the mean; and how many independent experiments were performed.
Specific remarks:
Fig. S1a, this figure certainly does not show that the ‘level of (Rac1) expression correlates with grade progression’. What is shown on the Y-axis, and what do the red and grey boxes represent? What is the source of these data, TCGA?
Fig. S1c, are the effects of 10 µM EHT1864 in the CCK-8 assay statistically different from the control cells? (how many independent experiments have been performed, are results shown the mean +/- SEM or SD?)
Fig. 2c, d, e: have the same number of control treated versus EHT1864-treated cells been scored?
Fig. 3b: it is very hard to see the stainings in the enlarged micrographs.
Fig. 4: please define how focal adhesions are discriminated from focal contacts. Please define how ‘adhesion assembly rate’ and ‘adhesion disassembly rate’ were measured. What is the difference between Fig. 4g and 4h? +/- EHT1864? Fig. 4i, FA lifetime measurements: why have so many more control cells been scored, compared to EHT1864-treated cells?
Fig. 6 e, j: stars indicate statistical significance, but the p-values are greater than 0.05.
Discussion and Conclusions: ‘Our results indicate that the upregulation of Rac1 is associated with an unusual motility leading to a higher degree of infiltration, which may be an attractive molecular target for GBM therapy… ‘In GBM cells, upregulation of Rac1 associate with a fast cell motility by regulating the dynamics of the cytoskeleton and cell adhesion.’
The authors do not investigate the effects of Rac1 upregulation on glioblastoma cell lines.
‘… cell mechanics have great potential for malignancy identification and classification.’’ ‘We suggest that cell mechanics can be used not only as a biomarker able to grade the metastatic state of cancer cells.’
It is unclear to this reviewer how exactly cell mechanics should be used for ‘malignancy identification and classification’. Has any attempt been made to correlate the observed mechanical properties with the well-established genomic markers used for glioma classification?
These are extremely general and far-reaching statements, which are not justified based on the presented data.
Fig. S1: Rac1 expression in patients with GBM compared with normal people. I would suggest to use a different term instead of ‘normal people’.
Fig. S3: Please clarify the experimental procedure. The very same cells are shown in the control group and in the EHT6814 panel. How have these images been taken, how has the experiment been conducted practically? How can the same cells be control treated for 0/15/30 min, and then incubated with EHT for the same time periods?
Fig. S4: what are 'tubulin assembly/disassembly fibers'?
Minor remarks:
At the end of the introduction, the authors state that they “established a stable mCherry-U87 cell line using 62 LV_Pgk1p-mCherry”, yet in the acknowledgements, they thank Prof. Antonello Mallamaci for the gift of this exact cell line. This statement is therefore misleading.
Glioblastoma cells invade, but they do not metastasize (see abstract, conclusions).
